# Uptake of iron and folic acid supplements among pregnant women in Dar es Salaam, Tanzania

Careen E. Koka[1], Saida Bakari[1], Belinda J. Njiro [2]*, Bruno F. Sunguya[1]

**1** School of Public Health and Social Sciences, Muhimbili University of Health and Allied Sciences, Dar es Salaam, Tanzania, **2** Department of Epidemiology and Biostatistics, School of Public Health and Social Sciences, Muhimbili University of Health and Allied Sciences, Dar es Salaam, Tanzania

* belindaj.njiro@gmail.com

## Abstract

Anemia in pregnancy is a public health concern globally with the highest prevalence observed in low and middle-income countries (LMICs). Tanzania is no exception. Iron and folic acid supplements (IFAS) intake is a proven intervention recommended to prevent anemia in pregnancy. Despite interventions in Tanzania, IFAS uptake has remained low due to reasons that are not well documented. This study aimed to assess the uptake and determinants of IFAS during pregnancy in Dar es Salaam, Tanzania. A quantitative cross-sectional study was conducted to assess levels and determinants of IFAS uptake during pregnancy in Dar es Salaam, Tanzania. Data was collected among 428 women post-delivery in postnatal wards of Temeke, Mwananyamala, and Amana regional referral hospitals. Descriptive analyses were conducted to determine levels and characteristics of IFAS uptake, whereas bivariate and multivariate analyses were conducted to examine determinants of uptake using SPSS version 23. Among the study participants, only 136 (31.8%) pregnant women who attended antenatal care (ANC) at referral hospitals had adequate IFAS uptake. Primiparous women were 74% less likely to achieve adequate IFAS uptake compared to multiparous women (aOR=0.26; 95% CI: 0.10 – 0.67). Women who attended five or more ANC visits (aOR=11.7; 95%CI: 1.30 – 63.66) and those from wealthier households (aOR=11.68; 95% CI: 2.91 – 89.57) were about 12 times more likely to achieve adequate IFAS uptake. Similarly, women from food-secure households had a ninefold higher likelihood of adequate uptake compared to those from severely food-insecure households (aOR=9.21; 95% CI: 1.82 – 10.33). Only one in three pregnant women attending regional referral hospitals in Dar es Salaam achieved adequate IFAS uptake to prevent anemia. Targeted interventions are urgently needed, particularly among pregnant women with higher parity, fewer ANC visits, and those facing economic hardship, food insecurity, and challenging family support.

**Data availability statement:** There are legal restrictions to sharing the datasets; we do not have approval to share data in public domains. However, data requests from individuals or entities shall be honored upon fulfilling the national regulatory requirements including data transfer agreement (DTA) as advised by the National Research and Ethics Committee (NIMR) of Tanzania. The datasets used and/or analyzed during the current study can be requested by contacting the Muhimbili University of Health and Allied Sciences (MUHAS) Directorate of Research and Publication at drp@muhas.ac.tz.

**Funding:** The author(s) received no specific funding for this work.

**Competing interests:** The authors have declared that no competing interests exist.

## Introduction

Anemia in pregnancy is diagnosed as hemoglobin concentration of less than 11.0 g/dl in the first trimester, 10.5g/dl in the second trimester and 11.0g/dl in the third trimester [1]. Iron deficiency is the most common cause of anemia in women of reproductive age and in pregnancy [1]. Iron is a crucial component of hemoglobin; a protein found in red blood cells (RBCs) that is responsible for carrying oxygen throughout the body. Human bodies need folic acid for the synthesis, repair, methylation, and cell division of DNA. Folic acid is also needed to develop healthy RBCs and to prevent anemia, especially during pregnancy and infancy [2].

Nutritional anemia is the most common type of anemia in pregnancy and is a significant public health challenge worldwide, particularly in low- and middle-income countries (LMICs) [3]. Anemia affects about 1.62 billion people globally, accounting for 24.8% of the world's population [4,5]. Pregnant women are the most vulnerable populations in nutritional anemia [6]; 56.4 million pregnant women are affected by anemia, and of them 17.2 million are from Africa [1]. Iron deficiency anemia, accounts for more than half of all occurrences of anemia in pregnancy, which is one of the major contributing factors to the worldwide burden of disease [7].

Pregnancy-related physiological changes, fetal growth, and development increase the need for iron and folic acid [8]. Due to the lower bioavailability of nutrients in pregnant women, the increased demand for essential nutrients cannot be met by diet alone [9]. If nutrition is not supplemented with iron and folic acid tablets throughout pregnancy, there is a considerable risk that iron deficiency and folic acid deficiency may manifest [9]. The World Health Organization (WHO) recommends iron and folic acid supplementation (IFAS) as one of the interventions for the control of anemia in pregnancy [1,7,10].

The WHO recommends that pregnant women take daily doses of 30–60 mg of iron and 0.4 mg of folic acid to prevent maternal anemia [11]. Iron and folic acid deficiency and anemia in pregnancy have been linked with maternal morbidity including fatigue, breathing difficulties, higher risk of postpartum depression, excessive blood loss during labor, and infections including puerperal sepsis. Negative consequences for the infants include low birth weight, neural tube defects such as spina bifida and anencephaly, intrauterine growth restriction, impaired fetal growth, preterm birth, and perinatal mortality [11–13]. The risk of all types of maternal anemia at term is reduced by 70% and iron deficiency anemia at term by 57%, if daily iron and folic acid supplements (IFAS) are taken throughout pregnancy [1]. The risk of infant mortality can also be decreased by 34% if a pregnant woman takes IFAS for 90 days during pregnancy [1]. Moreover, the levels of maternal folic acid and iron may also influence infants' intellectual development [14].

To attain a 50% reduction in anemia among women of reproductive age by 2025, the WHO recommends the provision of IFAS in pregnancy [2]. Reduced maternal and child morbidity and mortality are already notable with the use of IFAS [7]. However, the uptake has remained low in many regions and countries, including Tanzania. Evidence on the associated factors varies and sometimes lacking in some contexts including Tanzania. This study aimed to fill this evidence gap by documenting the uptake of IFAS and associated factors among pregnant women in Dar es Salaam, Tanzania

## Methodology

### Ethics statement

This study received ethical clearance from the Institutional Review Board of the Muhimbili University of Health and Allied Science (MUHAS IRB); ethical approval number MUHAS-REC-06-2024-2348. We sought the necessary permissions from the district administrative office and the respective hospitals at Temeke, Mwananyamala, and Amana regional referral hospitals. All participants provided written consent to participate before inclusion in the study. For participants aged less than 18 years, we obtained written assent; informed verbal consent was also obtained from their respective guardians. The information obtained from the participants was used for research purposes alone and are available to authorized personnel only.

### Study design and setting

This hospital-based cross-sectional quantitative study was conducted to assess the levels and determinants of IFAS uptake during pregnancy in Dar es Salaam, Tanzania. The study was conducted from 1$^{st}$ to 31$^{st}$ of July 2024 at three referral hospitals in Dar es Salaam, namely, Temeke, Mwananyamala and Amana regional referral hospitals located in Temeke, Ilala, and Kinondoni districts respectively.

### Study population

Data was collected from post-natal women attending the three health facilities at their respective post-natal wards. Both post-spontaneous vaginal delivery and post-cesarean section women that gave birth were eligible to participate and were involved in the study. All post-natal women aged 15–49 years visiting the post-natal wards at Temeke, Amana, and Mwananyamala regional referral hospitals were eligible to participate. Only women with a prior history of ANC visits to any health facility during their most recent pregnancy and who had their ANC cards for quality checks were included. The study excluded women who were in severe pain, had health emergencies and those who did not consent to participate in the study.

### Sampling and sample size

The hospitals were purposively selected owing to the volume of the clients. Participants were randomly selected for the study using a simple random sampling technique. The sample size was estimated by using Fisher's formula $N = Z^2 P (1-P)/E^2$, where N was the estimated minimum sample size; Z was the confidence level at 95% (standard value is 1.96); P was the proportion (prevalence of iron and folic acid adherence (50%); and E was the precision at 95% CI = 0.05. The minimum sample that was achieved for this study was 385 post-natal women. A 10% non-response rate was added to give a total sample size of 428 post-natal women. Participants were selected through a simple random sampling technique using a lottery technique whereby participants were assigned numbers each. Numbers were written on small pieces of paper, of the same size, color and shape. The papers were folded and then put in a box and then mixed. A blindfold selection was made. The process of selecting continued until the required sample size per respective hospital was reached. About 18 – 24 participants were selected in a day.

### Data collection and measurements

Data was collected from respondents via face-to-face interviews by the researcher and two trained research assistants using a semi-structured questionnaire adopted from the 2022 Tanzania Demographic Health Survey (TDHS) and previous studies [15,16]. All the questions were written in English and translated in Swahili (local language). The interviews lasted for approximately 30 minutes for each respondent; data collection was conducted at the respondents' respective postnatal wards. Data on maternal age, education level, employment status, and wealth index were collected. Weighted

wealth index was used as a proxy for household economic level. It considered the ownership of home resources, dwelling features, fuel for cooking and lighting, toilet type, water supply, and feeding habits which was used to determine economic status. The items were weighted and added together to create the wealth index. The weighted wealth index score was split similarly into three (tertiles) representing high, medium, and low economic levels. Maternal factors assessed included parity, timing of the first ANC, and number of ANC visits. Health facility factors included distance to the health facility and overall healthcare professional's attitude as reported by the respondents. Participants were asked to rank health care providers' attitude during their ANC visits as either good, fair, bad, and do not know.

The feeding frequency tool was adopted from previous Tanzanian research projects conducted in Tanzania [17]. The WHO suggests a feeding schedule of three meals and at the minimum two snacks in 24 hours for pregnant women [18]. The Dietary Diversity Scale (DDS) was used to measure dietary diversity adopted from previous studies conducted in Tanzania. A roster of twelve foods was extracted from the 2022 Tanzania Demographic Health Survey (TDHS) [15]. Participants were asked to mention the food items they had consumed on the previous day, and the final score for dietary diversity was determined. Dietary diversity below 4 was considered as low. Household food insecurity was assessed using a nine-item Household Food Insecurity Access Score (HFIAS) tool. On this scale, the options are: 0 = "no," 1 = "infrequently," 2 = "spoken" and 3 = "frequently". The total score ranges from 0 to 27. The food insecurity magnitude can be classified into four categories: highly insecure [18–27], severely insecure [11–17], mildly insecure [2–10], and secure (0 − 1).

The primary outcome for this study was the IFAS uptake of 90 pills or more. IFAS uptake was self-reported by the study respondents as the average number of days that participants took iron and folic acid supplements during their most recent pregnancy [15]. This information was double-checked on the Reproductive and Child Health (RCH) card used to record maternal and child assessments and progress during pregnancy and postpartum periods by the healthcare workers. In Tanzania, all pregnant women receive a single pill containing 200mg of ferrous sulphate and 0.4mg of folic acid daily [19]; this is equivalent to 65mg of elementary iron and 400 $\mu$g of folic acid daily starting at the first antenatal visit as WHO recommends [11]. WHO recommends a daily intake of iron and folic acid throughout the pregnancy [11]. Adequate IFAS uptake was defined as the intake of folic acid and iron tablets for a minimum of 90 days per pregnancy adopted from previous studies in similar settings from Tanzania and Pakistan and using TDHS methodologies [1,20,21]. IFAS uptake was categorized as normal uptake (90 or more pills per pregnancy) and low uptake (less than 90 pills per pregnancy); the adequate IFAS uptake in our study was equivalent to at least a third of the WHO recommended dose in pregnancy [20,21].

## Data analysis

The collected data was analyzed to identify the determinants of IFAS uptake using both descriptive and logistic regression analyses. Descriptive analysis was employed to summarize the study population characteristics. IFAS uptake was categorized as normal uptake (90 pills or more per pregnancy) and low uptake (less than 90 pills per pregnancy). IFAS uptake was defined as the proportion of respondents who reported IFAS intake of 90 pills or more during their recent pregnancy. The Pearson Chi-square test was used to examine the differences in the proportions of IFAS uptake by participants' characteristics. Bivariate and Multivariate logistic regression analyses were computed, and crude and adjusted odds ratios with corresponding 95% confidence intervals (95% CI) were recorded. Variables were included in the final model if having a p-value of <0.2 in the bivariate analysis. All analyses were conducted using IBM SPSS version 23.0. A p-value of <0.05 was considered statistically significant.

## Patient and public involvement

Patient or the public were not directly involved in the design and conduct of this research. However, as part of our dissemination plan, the final report from our analysis will be shared with the included health facilities to inform them of the status of IFAS uptake and the relevant recommendations.

## Results

### Participant's characteristics

A total of 428 participants (post-delivery women) were recruited for the study with full informed consent. **Table 1** shows the participants' characteristics stratified by the adequacy of IFAS uptake. Among all participants, only 31.8% (n = 136) had adequate IFAS uptake during pregnancy. The highest proportion of adequate IFAS uptake was among participants with university-level education (53.8%, n = 28), multiparous women (42.0%, n = 58), women who booked their ANC visit in the first trimester (53.5%, n = 106), and those that had five or more ANC visits. The proportion of adequate IFAS uptake was lower among women in the low wealth index and those who reported experiencing poor healthcare provider attitudes.

### Determinants of iron and folic acid uptake

Table 2 shows the bivariate and multiple logistic regression analysis findings. The odds of adequate IFAS uptake were 74% lower among primiparous women as compared to multiparous women (aOR=0.26; 95% CI: 0.10 – 0.67). Women who reported five or more ANC visits were 12 times more likely to have adequate IFAS uptake than those who attended 1–4 ANC visits (aOR=11.7; 95% CI: 1.62 – 84.34). Women from high-wealth index families were12 times more likely to report adequate IFAS uptake than women whose families had low wealth index (aOR=11.68; 95% CI: 2.91 – 89.57). Living in a food-secure household was also associated with higher IFAS uptake by nine times compared to women whose households had severe food insecurity (aOR=9.21; 95% CI: 1.82 – 10.33).

## Discussion

Less than one in three women had adequate of iron and folic acid supplementation during pregnancy in Dar es Salaam, Tanzania. Inadequate IFAS uptake in the context of the high burden of anemia among pregnant women calls for immediate and sustained efforts to address the burden of anemia in pregnancy. In Tanzania, similar results were documented in the 2022 TDHS, where the IFAS uptake for 90 days and more was reported at 40.8% [16]. These IFAS uptake findings are also not very different from evidence from a neighboring country, Kenya [22]. The current findings show an improvement compared to a previous study conducted in Moshi Tanzania in 2009, where 16.1% of women reported taking folic acid and iron supplements [23]. However, this study only reported on IFA supplements intake without reporting the adequacy of IFAS intake. The slight improvement reported could be attributed to improved ANC attendance, access to nutritional counseling at the ANC, increased number of health facilities and trained health workers, and campaigns on facility-based child deliveries by the government and implementing partners. It is worth noting that the adequacy of IFAS uptake in this study was low despite using a suboptimal cut-off point of 90 days/pills; this is only a third of what WHO recommends [11]. This highlights that the IFAS uptake is significantly lower than the recommended optimal uptake, immediate action is therefore needed to address this gap.

Parity was associated with the uptake of IFAS among pregnant women. Primiparous women were less likely to have adequate IFAS uptake compared to multiparous women. Similar findings have been reported in Tanzania, Uganda and other similar contexts [20,24,25]. Multiple pregnancies might have increased exposure to health education provided to pregnant women in ANC, leading to a better understanding of the risk of anemia and its complications and more experience and tolerance to iron supplement pills. These might have increased the uptake among those with a previous history of pregnancies.

The number of ANC visits also influenced the uptake of IFAS among pregnant women in this study. The more frequently a pregnant woman attends the clinic, the more she is exposed to frequent interactions with healthcare workers, and the more she is followed up on IFAS uptake. This increases their access to iron and folic acid supplementation, and more chances to receive nutritional education. In each visit, a woman would be tested for hemoglobin, and this may be an indicator of the increased need to take IFAS. Similar evidence was reported in a systematic review and meta-analysis

PLOS Global Public Health

**Table 1. Uptake of IFAS by participants' characteristics (N = 428).**

| Variables | Total n (%) | Adequate uptake, n (%) | Inadequate uptake, n (%) | p-value |
|---|---|---|---|---|
| **Age groups** | | | | 0.688 |
| 16 – 25 | 188 (43.9) | 56 (41.2) | 132 (45.2) | |
| 26 – 35 | 194 (45.3) | 62 (45.5) | 132 (45.2) | |
| 36 – 45 | 46 (10.7) | 18 (13.2) | 28 (9.6) | |
| **Highest education level** | | | | |
| Primary level | 142 (33.2) | 52 (34.7) | 90 (33.1) | 0.027 |
| Secondary level | 228 (53.3) | 70 (46.7) | 158(58.1) | |
| University | 52 (12.1) | 28 (18.7) | 24 (8.8) | |
| **Parity** | | | | |
| Primiparity | 164 (38.3) | 38 (27.9) | 126 (43.2) | 0.046 |
| Secundiparity | 126 (29.4) | 40 (29.4) | 86(29.5) | |
| Multiparity | 138 (32.2) | 58 (42.6) | 80(27.4) | |
| **Time of first booking** | | | | |
| 1st trimester | 198 (46.3) | 106 (70.7) | 92 (33.8) | 0.003 |
| 2nd& 3rdtrimester | 200 (46.7) | 44 (28.2) | 156(57.4) | |
| Don't know | 30 (7.0) | 6 (8.8) | 24 (8.8) | |
| **Number of ANC visits** | | | | |
| 1 – 4 | 168 (39.3) | 24 (17.6) | 144 (49.3) | <0.001 |
| 5 and above | 212 (49.5) | 104 (76.5) | 108 (37.0) | |
| Don't know | 48 (11.2) | 8 (5.9) | 40 (13.7) | |
| **Wealth index** | | | | |
| High | 166 (38.8) | 74 (54.4) | 92 (31.5) | 0.001 |
| Middle | 173 (40.4) | 55 (40.4) | 118 (40.4) | |
| Low | 89 (20.8) | 7 (5.1) | 82 (28.1) | |
| **Distance to health facility affecting treatment** | | | | |
| Yes | 158 (36.9) | 36 (26.5) | 122 (41.8) | 0.031 |
| No | 270 (63.1) | 100 (73.5) | 170 (58.2) | |
| **Healthcare worker's attitude** | | | | |
| Good | 292 (68.2) | 106 (77.9) | 186 (63.7) | 0.037 |
| Fair/bad | 136 (31.8) | 30 (22.1) | 106 (36.3) | |
| **Household food security** | | | | 0.011 |
| Food secure | 226 (52.8) | 95 (69.9) | 131 (44.9) | |
| Mildly insecure | 30 (7.0) | 11 (8.1) | 19 (6.5) | |
| Moderately insecure | 77 (18.0) | 15 (11.0) | 62 (21.2) | |
| Severely insecure | 95 (22.2) | 15 (11.0) | 80 (27.4) | |
| **Dietary diversity score** | | | | 0.159 |
| Below 4 | 208 (48.6) | 55 (40.4) | 153 (52.4) | |
| 4 and above | 220 (51.4) | 81 (59.6) | 139 (47.6) | |
| **Feeding frequency** | | | | 0.002 |
| Low | 143 (33.4) | 22 (16.2) | 121 (41.4) | |
| Good | 285 (66.6) | 114 (83.8) | 171 (58.6) | |

**Table 2. Bivariate and multivariate analysis of determinants of IFAS uptake.**

| Variable | Bivariate analysis | | Multivariate analysis | |
|---|---|---|---|---|
| | cOR (95%CI) | *p value* | aOR (95%CI) | *p value* |
| **Highest education level** | | | | |
| Primary | 0.29 (0.11 – 0.74) | 0.010 | 0.54 (0.16 – 1.91) | 0.341 |
| Secondary | 0.38 (0.16 – 0.90) | 0.029 | 0.65 (0.20 – 2.11) | 0.474 |
| University | Reference | | Reference | |
| **Parity** | | | | |
| Primiparity | 0.42 (0.21 – 0.84) | 0.014 | 0.26 (0.10 – 0.67) | 0.005 |
| Secundiparity | 0.64 (0.31 – 1.31) | 0.223 | 0.58 (0.23 – 1.47) | 0.246 |
| Multiparity | Reference | | Reference | |
| **Time of first booking** | | | | |
| 1st trimester | 3.07(0.82 – 11.57) | 0.097 | 0.82 (0.09 – 7.19) | 0.857 |
| 2nd & 3rd trimester | 1.13 (0.29 – 4.26) | 0.861 | 0.54 (0.06 – 4.95) | 0.585 |
| Don't know | Reference | | Reference | |
| **Number of ANC visits** | | | | |
| 1 – 4 | 0.83 (0.24 – 2.87) | 0.772 | 2.23 (0.29 – 16.86) | 0.439 |
| 5 and above | 4.82 (1.54 – 15.04) | 0.007 | 11.70 (1.30 – 63.66) | 0.015 |
| Don't know | Reference | | Reference | |
| **Wealth index** | | | | |
| High | 16.25 (3.0 – 88.9) | 0.001 | 11.68 (2.91 – 89.57) | 0.059 |
| Middle | 6.96 (1.3 – 36.5) | 0.022 | 6.59 (0.69–62.96) | 0.102 |
| Low | Reference | | Reference | |
| **Distance to health facility** | | | | |
| Yes | 1.99 (1.06 – 3.75) | 0.032 | 1.92 (0.82 – 4.51) | 0.135 |
| No | Reference | | Reference | |
| **Health care workers attitude** | | | | |
| Good | 0.50 (0.26 – 0.97) | 0.039 | 0.62 (0.25 – 1.56) | 0.313 |
| Fair/bad | Reference | | Reference | |
| **Household food security** | | | | |
| Food secure | 6.50 (1.73 – 24.39) | 0.006 | 9.21 (1.82 – 10.33) | 0.047 |
| Mild food insecurity | 4.50 (0.54 – 37.38) | 0.164 | 8.96 (0.16 – 13.13) | 0.234 |
| Moderate food insecurity | 1.33 (0.26 – 6.83) | 0.730 | 4.15 (0.47 – 6.24) | 0.820 |
| Severe food insecurity | Reference | | Reference | |
| **Dietary diversity score** | | | | |
| <4 (low) | 1.96 (0.77 – 4.99) | 0.161 | 0.73 (0.21 – 2.51) | 0.617 |
| 4 and above (good) | Reference | | Reference | |
| **Food frequency** | | | | |
| Low | 5.47 (1.83 –16.39) | 0.002 | 1.75 (0.31 – 9.82) | 0.526 |
| Good | Reference | | Reference | |

aOR: adjusted odds ratio, cOR: crude odds ratio, ANC: antenatal care

study done in Sub-Saharan Africa in 2021 [26]. Other studies also supported these findings [20,27]. Closer distance to the facility, knowledge of the importance of ANC visits, and good healthcare workers' attitudes also influence the frequency of ANC visits among pregnant women and hence more opportunities for IFAS [28,29].

Economic status as the key social determinant of health has an effect on the general population's health, as well as among pregnant women. In this study, the wealth index was also a determinant of IFAS uptake among pregnant women; those from wealthier households were more likely to have adequate uptake. Higher socioeconomic status is often connected to better education, greater access to health information, and more effective use of health services [30]. Moreover, socio-economic disadvantages, including poverty, negatively influence the uptake of many health interventions, including preventive interventions such as IFAS. As reported previously, women with higher incomes may have had better access to supplements due to their stronger financial ability to purchase despite being available for free in most ANC clinics [21,23,30,31].

Women whose households were food secure were more likely to have had adequate IFAS during pregnancy compared to those whose families had severe food insecurity. Women with food insecurity and low economic status may have other priorities, causing less physical access to ANC services. This may include engagement with economic activities and, hence, lack of time and financial means to attend ANC visits and obtain IFAS. Due to this, they may also lack access to education on the importance of supplementation. Previous evidence shows a disproportionally higher burden of iron deficiency anemia among food-insecure pregnant women [32]. Women from households with food insecurity have minimum or no intake of iron and folic acid supplements and may be faced with additional challenges of lacking iron and folic acid-rich foods.

This study reports an update on the key predictors of IFAS supplementation in urban Tanzania. These findings should be interpreted in the light of the following limitations. Firstly, there was a potential for recall bias as participants were required to remember the days on which they had taken IFAS tablets during pregnancy and the ANC visits they had attended. We mitigated this by comparing such reports with the alternative source (the RCH card) which partly confirmed if the participants attended ANC to receive IFAS as recommended. Secondly, some of the study participants did not respond to some of the questions hence this study has some missingness which was mitigated by including the non-response rate in sample size calculation. Lastly, there might be social desirability bias since the questions were asked by the researchers (a doctor and a nurse). This might have influenced a positive response by the study participants. This was mitigated by guaranteeing total confidentiality to the study participants and allowing anonymity by using numbers and leaving out any identifiable information. Given that our study was conducted in urban health facilities, the generalizability of our findings may not apply in rural settings.

## Conclusion

The uptake of iron and folic acid supplements among pregnant women is still low, only 31.8% of pregnant women had IFAS uptake for 90 days or more during their most recent pregnancy. Addressing this severe challenge in the uptake of life-serving intervention among the identified vulnerable population with a high burden of anemia in Tanzania, efforts should be focused on primiparous women, women with a low number of ANC visits, and households with food insecurity and low wealth quintiles. Since this study focused only on women who attended at least one ANC visit, we recommend further studies and interventions to address IFAS uptake for pregnant women with poor or no ANC access.

## Acknowledgments

The authors would like to thank the medical officers in charge, the administrative office at the Temeke, Mwananyamala, and Amana regional referral hospitals and all study participants.

## Author contributions

**Conceptualization:** Careen E. Koka, Saida Bakari, Bruno F. Sunguya.

**Data curation:** Careen E. Koka, Belinda J Njiro.

**Formal analysis:** Careen E. Koka.

**Methodology:** Careen E. Koka, Saida Bakari, Belinda J Njiro, Bruno F. Sunguya.

**Supervision:** Saida Bakari, Bruno F. Sunguya.

**Validation:** Saida Bakari, Belinda J Njiro, Bruno F. Sunguya.

**Visualization:** Careen E. Koka.

**Writing – original draft:** Careen E. Koka.

**Writing – review & editing:** Saida Bakari, Belinda J Njiro, Bruno F. Sunguya.

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
