## [Decision Letter · Decision Letter 0]

3 Apr 2025

PGPH-D-25-00011

Iron and folic acid supplements uptake among pregnant women in Dar es Salaam, Tanzania

Dear Dr. Njiro,

Thank you for submitting your manuscript to PLOS Global Public Health. After careful consideration, we feel that it has merit but does not fully meet PLOS Global Public Health’s publication criteria as it currently stands. Therefore, we invite you to submit a revised version of the manuscript that addresses the points raised during the review process.

Please note that we have only been able to secure a single reviewer to assess your manuscript. We are issuing a decision on your manuscript at this point to prevent further delays in the evaluation of your manuscript. Please be aware that the editor who handles your revised manuscript might find it necessary to invite additional reviewers to assess this work once the revised manuscript is submitted. However, we will aim to proceed on the basis of this single review if possible. 

The manuscript has been evaluated by one reviewer, and their comments are available below.

The reviewers have raised a number of concerns that need attention. Could you please revise the manuscript to carefully address the concerns raised?

We look forward to receiving your revised manuscript.

Kind regards,

Johanna Pruller, Ph.D.

Staff Editor

Journal Requirements:

Additional Editor Comments (if provided):

Reviewers' comments:

**Comments to the Author**

1. Does this manuscript meet PLOS Global Public Health’s publication criteria?

Reviewer #1: Yes

2. Has the statistical analysis been performed appropriately and rigorously?

Reviewer #1: No

3. Have the authors made all data underlying the findings in their manuscript fully available (please refer to the Data Availability Statement at the start of the manuscript PDF file)?

Reviewer #1: Yes

4. Is the manuscript presented in an intelligible fashion and written in standard English?

Reviewer #1: Yes

Reviewer #1: This manuscript describes a cross-sectional study to assess self-reported IFA consumption during pregnancy in women admitted to three referral hospitals in Dar es Salaam, Tanzania.

The manuscript is nicely organized and generally well written, but I have a few concerns with how the data is presented and have a few suggestions to improve the clarity of the reported findings:

Is there specific medical reasons why women are sent to referral hospitals in this region? Are the pregnant individuals different from those that would normally visit non-referral hospitals? Curious if this would influence the generalizability of the results?

Data in tables is presented oddly. Each percentage reported is for each line of the variable, which is mis-leading. Suggest for it to be switched so that percentages reflect each variable as grouped, as a total of 100% represented vertically (rather than horizontally). Also there are percentages and other data missing from some categories, and suggest to be consistent in either presenting whole numbers or numbers with decimal places for each percent reported.

Why not consider IFA uptake as a continuous number, to have more power to detect associations across the measured variables/predictors?

Line 141: 90 pills or more. Why is the threshold of 90 pills used in this study. Is this what the MoH recommends in Tanzania? More detail could be provided on this. How many pills should women be receiving?

Line 144: It is not clear how the RCH card was used to ‘verify’ the number of IFA tablets consumed. Was it just written in as a number by women or health care providers?

Line 147: 200 mg ferrous sulphate should correspond to an exact content of elemental iron, not a range of 30-60 mg. Please check and correct.

Line 153-4: “adequate IFA uptake is defined as four IFA tablets every week” – who defined this? Is there a reference that could be included here for context?

What about women with severe anemia – did they receive any other intervention or suggested dosing regime (e.g. supplements with a higher iron content)? Or IV infusions?

Line 99: why did you only include women with a prior history of ANC? How many women were excluded for these exact reasons? This would directly influence the generalizability of your results especially if you excluded the most vulnerable people with poor access (equity)

Acceptability is a very important consideration and factor as to why pregnant individuals may not be consuming the IFA. Was this assessed in the questionnaire?

Line 53: thresholds for anemia during pregnancy actually have different trimester-specific thresholds. Please see the latest 2024 WHO guidance on thresholds for anemia in individuals and populations.

Line 55: “significantly dependent on iron” – iron is a component of the RBC, suggest to rephrase this text.

Line 66: “due to the lower bioavailability of nutrients in pregnant women” please add a reference for this statement.

Line 69: “… there is a considerable risk that iron deficiency and folic acid deficiency may manifest” – in fact, there are much more important consequences for both mother and infant that could be referenced here.

Line 118: TDHS – which one?

Line 211: Explain the results reported in the Mbeya region (rather than just saying they were ‘similar’)

Line 213: reference to Moshi study – did they assess IFA in the similar approach, considering 90 tablets as the threshold for adequate intake? Was it self reported?

**Do you want your identity to be public for this peer review?** For information about this choice, including consent withdrawal, please see our Privacy Policy

Reviewer #1: No

---

## [Decision Letter · Decision Letter 1]

14 Jul 2025

PGPH-D-25-00011R1

Iron and folic acid supplements uptake among pregnant women in Dar es Salaam, Tanzania

Dear Dr. Njiro,

Thank you for submitting your manuscript to PLOS Global Public Health. After careful consideration, we feel that it has merit but does not fully meet PLOS Global Public Health’s publication criteria as it currently stands. Therefore, we invite you to submit a revised version of the manuscript that addresses the points raised during the review process.

Your manuscript has been assessed by two reviewers, including Reviewer #1 from the previous round of review. Both reviewers demonstrate interest in your work, however, they raise comments for you to address. We recommend considering these points careful when preparing your revised manuscript and response-to-reviewers document.

We look forward to receiving your revised manuscript.

Kind regards,

Dr Jason Morgan

Staff Editor

Journal Requirements:

Additional Editor Comments (if provided):

Reviewers' comments:

Reviewer's Responses to Questions

**Comments to the Author**

Reviewer #1: All comments have been addressed

Reviewer #2: All comments have been addressed

publication criteria?

Reviewer #1: Yes

Reviewer #2: Yes

3. Has the statistical analysis been performed appropriately and rigorously?

Reviewer #1: Yes

Reviewer #2: Yes

4. Have the authors made all data underlying the findings in their manuscript fully available (please refer to the Data Availability Statement at the start of the manuscript PDF file)?

Reviewer #1: Yes

Reviewer #2: Yes

5. Is the manuscript presented in an intelligible fashion and written in standard English?

Reviewer #1: No

Reviewer #2: Yes

Reviewer #1: Authors have provided sufficient responses to my initial queries; however, a few additional edits should be made:

1. Line 25 of the abstract: suggest to revise 'biggest brunt' to 'highest prevalence observed'

2. Suggest to use the acronym IFA rather than IFAS, especially as in the abstract the word 'supplement' is not included at first mention. IFA is more common.

3. Consider slight rewording of title to: "Uptake of IFA..." rather than IFA uptake..."

4. Line 31 suggest rewording to 'women post-delivery' rather

5. Line 35 - instead of "one in three" please report the actual observed n= from the study

6. Line 55 - anemia does not need to be capitalized

7. Several acronyms are used throughout but not consistently. Check line 109 'ANC' and check line 93 'IFA' and also comprehensively check all acronyms with a search and replace to ensure consistency throughout.

8. Line 233 - reference to the WHO recommendation should include a citation

9. As this journal does not have a copy editor, strongly suggest that the grammar be reviewed before the final version is submitted.

Reviewer #2: The manuscript has very minor corrections.

**Do you want your identity to be public for this peer review?** For information about this choice, including consent withdrawal, please see our Privacy Policy

Reviewer #1: No

Reviewer #2: **Yes: ** Dr. Eunice Njogu

---

## [Decision Letter · Decision Letter 2]

18 Sep 2025

Uptake of iron and folic acid supplements among pregnant women in Dar es Salaam, Tanzania

PGPH-D-25-00011R2

Dear Dr. Njiro,

We are pleased to inform you that your manuscript 'Uptake of iron and folic acid supplements among pregnant women in Dar es Salaam, Tanzania' has been provisionally accepted for publication in PLOS Global Public Health.

Best regards,

Julia Robinson

Executive Editor

Reviewer #2:

Reviewer Comments (if any, and for reference):

Reviewer's Responses to Questions

**Comments to the Author**

Reviewer #2: All comments have been addressed

publication criteria?

Reviewer #2: Yes

3. Has the statistical analysis been performed appropriately and rigorously?

Reviewer #2: Yes

4. Have the authors made all data underlying the findings in their manuscript fully available (please refer to the Data Availability Statement at the start of the manuscript PDF file)?

Reviewer #2: Yes

5. Is the manuscript presented in an intelligible fashion and written in standard English?

Reviewer #2: Yes

Reviewer #2: The manuscript is in good form and content, All the comments have been addressed and the manuscript is ready for publication.

**Do you want your identity to be public for this peer review?** For information about this choice, including consent withdrawal, please see our Privacy Policy

Reviewer #2: **Yes: ** Dr. Eunice Njogu
